# Biotechnological Test of Plant Growth-Promoting Bacteria Strains for Synthesis of Valorized Wastewater as Biofertilizer for Silvicultural Production of Holm Oak (*Quercus ilex* L.)

**DOI:** 10.3390/plants14172654

**Published:** 2025-08-26

**Authors:** Vanesa M. Fernández-Pastrana, Daniel González-Reguero, Marina Robas-Mora, Diana Penalba-Iglesias, Pablo Alonso-Torreiro, Agustín Probanza, Pedro A. Jiménez-Gómez

**Affiliations:** Department of Pharmaceutical Science and Health, CEU San Pablo University, Montepríncipe Campus, Ctra. Boadilla del Monte Km 5.300, 28668 Boadilla del Monte, Madrid, Spain; vanesa.fernandezpastrana@ceu.es (V.M.F.-P.); diana.penalbaiglesias@ceu.es (D.P.-I.); pablo.alonsotorreiro@usp.ceu.es (P.A.-T.); a.probanza@ceu.es (A.P.); pedro.jimenezgomez@ceu.es (P.A.J.-G.)

**Keywords:** biofertilizer, WWTP, PGPB, *Bacillus pretiosus*, *Pseudomonas agronomica*, *Quercus ilex*, metagenomics, antibiotic sensitivity

## Abstract

The degradation of Mediterranean forest ecosystems, such as holm oak forests, has intensified in recent decades due to climate change, forest fires, and deforestation, compromising the natural regeneration of the soil. In this context, it is essential to apply sustainable strategies to restore soil and promote plant growth, thus helping the regeneration of the ecosystem. One of these strategies is the use of plant growth-promoting bacteria (PGPB) in combination with recovered organic waste, such as that from wastewater treatment plants (WWTPs). In this paper, the effects of a biofertilizer formulated from WWTP residue (with and without sterilization), supplemented with two PGPB strains (*Bacillus pretiosus* and *Pseudomonas agronomica*), on the growth of holm oak seedlings (*Quercus ilex*) were evaluated under field conditions. A study was carried out on its nutritional composition, the rhizospheric cenoantibiogram, and its functional and taxonomic microbial diversity. Nine combinations of chemical and biological treatments using irrigation with water as a control were compared. The results showed that treatments with WWTP, especially combined with PGPB strains, promoted greater plant development and a lower seedling mortality rate. The cenoantibiogram exhibited a reduction in the resistance profile in soils treated with biofertilizer, without affecting soil microbial diversity, which remained unaltered across treatments, as confirmed by metagenomic and functional diversity analyses. Overall, this research reinforces the viability of the use of biofertilizers recovered from WWTP as an ecological and effective strategy for the recovery of degraded holm oak forests.

## 1. Introduction

The holm oak (*Quercus ilex* L.) is one of the most emblematic tree species of the Mediterranean ecosystem. Its presence extends throughout the Mediterranean basin, where limestone and arid soils prevail, adapting efficiently to prolonged drought conditions [1]. This adaptation ability is due to its characteristic morphology, which includes a deep root system enabling sustained groundwater access [2]. It also plays a key role in soil conservation by preventing erosion, stabilizing the substrate, and improving water infiltration, being crucial to maintaining the sustainability of holm oak and pasture forests [3]. Furthermore, it shows remarkable physiological plasticity in the use of water and nutrients, which allows its adaptation to various environmental conditions [4]. The holm oak is an evergreen tree. Its leaves are hard, leathery (sclerophyllous), and oval-shaped, with often toothed margins. The adaxial surface of the leaves is dark green and glossy, whereas the abaxial surface is whitish due to the presence of trichomes, which contribute to reducing transpirational water loss. Leaf morphology can vary within the same tree by adapting to local conditions. The trunk of the holm oak is robust and can reach a considerable diameter. The bark of juvenile holm oaks is smooth and grayish, but as trees mature, it develops deeper fissures and a rougher texture, a trait that improves resilience to heat stress and fire damage.

Its wood has traditionally been used in the charcoal and firewood industry and even in industries such as construction and cabinetmaking [2]. Holm oak forests face major pressures from overexploitation, unsustainable land-use practices, and the rising frequency of forest fires. Climate change further compounds these drivers, with projected warming and aridification expected to significantly curtail their distribution and regeneration capacity [5,6,7].

Fires pose a global threat to forest ecosystems, but in Mediterranean landscapes, their impact is exacerbated by ecosystem fragility and the slow regeneration of woody vegetation. In addition to the visible loss of trees and biodiversity, fires also cause changes in the cycles of carbon and nutrients in the soil. Recent studies have revealed that these fires significantly modify the microbial communities of the soil [8,9,10]. For this reason, experts insist on a commitment to a more complete ecological restoration, which also includes the recovery of the soil and the microorganisms that inhabit it [11]. Soil microbial communities play an essential role in the health and stability of ecosystems. However, in many forest restoration strategies, the recovery of these communities has been pushed into the background. Modern restoration techniques have begun to integrate the use of organic amendments and biofertilizers to encourage the recovery of the soil microbiota [12].

The use of biofertilizers based on plant growth-promoting bacteria (PGPB) has established itself as an innovative strategy to improve soil structure and increase nutrient availability, especially in soils degraded by fires or intensive agricultural practices [13]. PGPBs, among which the genera *Bacillus* and *Pseudomonas* stand out, have been shown to be effective both in phosphorus solubilization and nitrogen fixation, as well as in the production of phytohormones, improving the resistance of plants to abiotic stress conditions such as drought, salinity, and high temperatures [14,15]. These PGPB are well known to produce phytohormones such as 3-indolacetic acid, which promotes radicular elongation, produce siderophores to improve the uptake, solubilize phosphates improving growth ratios, and fix nitrogen or degrade complex organic substances for an easier uptake for the plants [14,15].

At the same time, the circular economy and agricultural sustainability approach has promoted the recovery of organic waste, such as sludge from wastewater treatment plants (WWTPs). Spain produces approximately 1.2 million tonnes of this waste annually, and its disposal remains a significant environmental challenge [16]. Taking advantage of WWTP sludge to produce biofertilizers not only represents a solution to reduce their environmental impact but also opens new opportunities to improve soil quality [17]. However, it is essential to ensure that the bacterial strains used in the production of these biofertilizers are free of virulence genes or antibiotic resistance, to prevent their spread in the environment [18]. Given that field outcomes frequently diverge from results obtained under controlled settings, recent studies stress the importance of conducting field trials to corroborate experimental findings [19]. Therefore, the evaluation of the safety and effectiveness of biofertilizers should focus on agronomic aspects and be aligned with broader principles of public and environmental health, such as those proposed by the ‘One Health’ approach. This approach promotes a holistic view of health, recognizing the interconnectedness between human, animal, and environmental health [20].

Combining various biotechnological approaches, such as the use of biofertilizers and metagenomics, with more traditional ecological restoration strategies offers a promising prospect for the rehabilitation of degraded soils. Recent papers have exhibited that the reintroduction of PGPBs adapted to the local environment promotes faster recovery of vegetation and improves the resilience of ecosystems [21].

It is necessary to previously analyze the possible risks that the strain or strains may have in the ecosystem [22]. For this purpose, the strains must be subjected to an analysis of resistance to antibiotics such as the cenoantibiogram. These techniques are essential prior to the release of bacteria into the environment, to avoid causing damage to both the ecosystem and human health. For all these reasons, a “One Health” approach must always be considered.

Community cohesion must also be considered, since the introduction of exogenous strains should preserve the functional diversity of the native microbiome. Metabolic profiling, for instance through Biolog EcoPlates, allows for comparative analyses between inoculated and baseline communities [23].

It is essential to carry out a global analysis of the microbial community in a metagenomic study where it is intended to introduce bacteria exogenously. The incorporation of new organisms can cause alterations in the stability of the community. A metagenomic approach allows the analysis of all microorganisms present in the soil. For this purpose, amplicon sequencing, specifically the *16S rRNA gene*, is used, which allows the characterization of the present bacterial populations. In addition, α and β diversity analyses need to be applied to assess genetic variability within and between microbial communities [24]. This approach is part of what is known as environmental genomics, ecogenomics, or community genomics [25].

Given the increasing need to restore and protect holm oak forests and to rehabilitate abandoned or degraded soils, the present study investigates the growth response of *Q. ilex* under natural field conditions. Plant growth promotion and nutritional composition were evaluated after inoculation with two PGPB strains (*Bacillus pretiosus* and *Pseudomonas agronomica*). The strains were delivered using organic waste recovered from a wastewater treatment plant, applied either in its raw form (EDAR) or after sterilization (EDARST). In addition, the impact of inoculation on soil antibiotic resistance was assessed through community-level assays (cenoantibiograms), with emphasis on resistance reduction and potential transmissibility. Rhizospheric responses were further characterized by phenotypic (Biolog ECO) and metagenomic analyses to determine the persistence and functional integration of the introduced strains within the *Q. ilex* microbiome.

## 2. Results

### 2.1. Biometry

The biometry analysis aimed to show how the different treatments affect the growth of *Q. ilex,* where the results (Figure 1) show an improvement due to the treatments.

Figure 1 exhibits a general trend of increased dry weight and aerial length in plants treated with bacterial strains, particularly under EDAR and EDARST conditions. In terms of dry weight, increments relative to the water control (WC0) were 16.5%, 19.2%, 48.4%, 46.2%, 49.4% and 37.4% for WC1, WC2, EDARC1, EDARC2, EDARSTC1 and EDARSTC2, respectively. For stem length, increases relative to WC0 were 54.2%, 24.7%, 25.3% and 21.8% for EDARC1, EDARC2, EDARSTC1 and EDARSTC2, respectively. Among the inoculated treatments, both *B. pretiosus* (C1) and *P. agronomica* (C2) exhibited comparable improvements, particularly under EDAR irrigation. In this treatment, sterilization eliminates the remaining background microbiota, thereby allowing the exclusive effects of the two inoculated strains to be observed. These findings support that both strains exert a similar and beneficial influence on plant growth under conditions of high organic load.

Table 1 shows that the mortality rate after 1 year of growth in a natural environment decreases when EDAR treatments are used, both in the application of the residue alone and in combination with the C1 and C2 strains.

### 2.2. Nutritional Analysis

#### Major Components (PCA), by Treatment

As presented in Figure 2, biological treatments with PGPB (plant growth-promoting bacteria), *Bacillus pretiosus* (C1) and *Pseudomonas agronomica* (C2), are segregated when compared to controls. This indicates that these treatments modify the functional profile of the plants in a consistent and different way from the control (C0), particularly when irrigation with treated wastewater (EDAR and EDARST) is used.

### 2.3. Functional Diversity: Biolog Ecoplate

An analysis of the functional profile by nutrient families was performed. This representation allows for evaluating the variation in nutrient consumption due to rhizospheric microbial communities.

As shown in Figure 3, differentiated patterns can be seen according to the irrigation treatment, highlighting a higher consumption of carbohydrates in the EDAR treatments added with the strains. Likewise, the reduction observed in amino acid consumption in treatments with EDAR and EDARST could suggest an improvement in the availability of mineral nitrogen. Moreover, a higher consumption of carboxylic acids is observed in soils treated with EDARST together with C1 (*B. pretiosus*). Regarding amine consumption, a slight change was detected between treatments, although without a clear pattern of functional differentiation associated with the type of irrigation, chemical treatment (TQ), and biological treatment (TB), which indicates that its use is not differential in the separation between communities. On the other hand, the use of phenolic compounds remains relatively homogeneous between treatments, suggesting that they do not represent a critical variable in functional differentiation either.

**Figure 2 plants-14-02654-f002:**
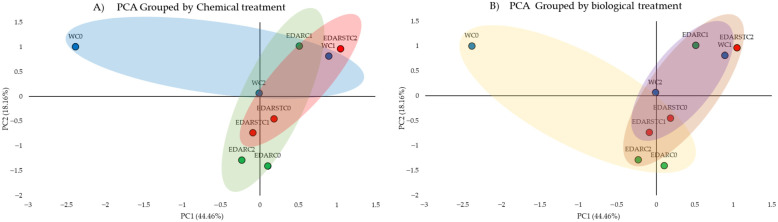
Multivariate analysis of principal components (PC1: 44.46%, PC2: 18.16%) showing the overall effect of treatments on the functional profile of the holm oak. Ellipses indicate grouping by type of irrigation (**A**) and type of biotreatment (**B**).

**Figure 3 plants-14-02654-f003:**
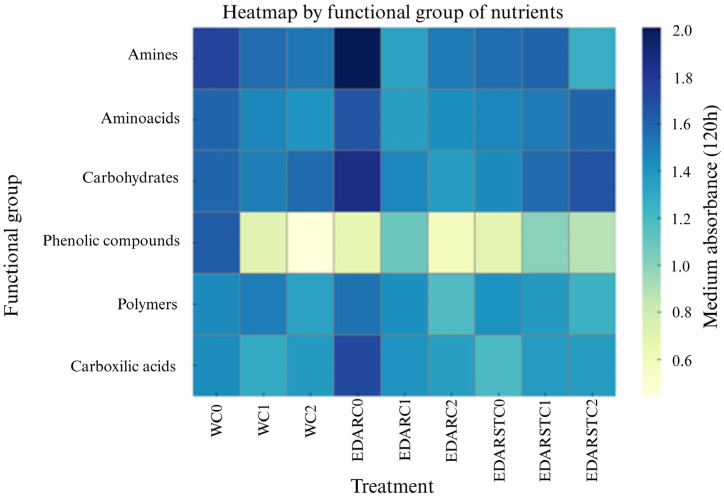
Heatmap containing absorbance values at 120 h of incubation for 6 key functional groups: amines, amino acids, carbohydrates, phenolic compounds, polymers, and carboxylic acids. The color goes from yellow to deep blue from low degradation to high degradation ratio.

### 2.4. Cenoantibiogram

Prior to conducting the principal component analysis (PCA), a multivariate analysis of variance (MANOVA) was performed to assess whether resistance profiles varied according to the type of chemical treatment (TQ), biological treatment (TB), or their combination. The results indicated that TQ exerted a clear effect, while TB also showed a potential influence. Consequently, PCA was applied to provide a visual representation of how treatments clustered based on their resistance profiles.

#### 2.4.1. PCA

PCA analysis (Figure 4) shows that both the irrigation type and the application of bacterial strains clearly influence soil hardiness profiles. Soils irrigated with water behave differently, while those treated with EDAR are more homogeneous, and those that receive EDARST show lower resistance, which may be due to the lack of microbiota in the waste.

#### 2.4.2. Resistance Index

Figure 5 exhibits how the resistance of the soil varies according to the treatment applied. In general, soils without inoculation have a higher resistance load, especially in the case of water control (WC0). The introduction of bacterial strains significantly reduces this resistance, both in soils without residues and in those treated with WWTPs. The effect is even more evident in sterilized residue (EDARST), where the C1 strain achieves the lowest value of resistance.

### 2.5. Taxonomic Diversity (Alpha Diversity)

In total, 237,731 valid readings were obtained after the sequence trimming process (truncation to 240 bp in forward and reverse readings; trim-left = 0), with an average of 26,414 clean readings per sample (Appendix A). Shannon index rarefaction curves (Appendix A) reached a plateau before the selected clipping threshold, indicating that the sequencing depth was sufficient to capture the microbial diversity present in all samples and allowing for consistent comparisons between treatments. QC of raw sequences is given in Appendix A. No sample was excluded due to low number of readings.

The alpha diversity analysis, shown in Figure 6 and Table 2, was estimated using the Shannon entropy index and did not reveal statistically significant differences between the treatments, both chemical and biological, according to the Mann–Whitney test (*p* > 0.05; exact values, test statistics and sample sizes in Table 2). This suggests that the bacterial richness and bacterial ratio of the bacterial community were not substantially altered by the addition of biofertilizers (PGPB). The Raincloud plots (Figure 6) reflect the distribution of diversity values per treatment, showing a general consistency in the internal diversity of the samples, with similar medians between groups.

### 2.6. Beta Diversity

The PCoA of beta diversity shows a slight segregation by chemical/biological treatments (Figure 7). Beta diversity, analyzed using a PCoA based on weighted distances from UniFrac, showed partial separation between groups treated with different types of fertilizers, although this difference was not statistically significant.

The PERMANOVA (Adonis) test revealed that the type of fertilizer explained 30.6% of the observed variation in microbial community structure (R^2^ = 0.306, *p* = 0.060), while inoculated bacteria explained 25.4% (R^2^ = 0.254, *p* = 0.233). Overall, the results of PERMANOVA and PERMDISP (Appendix A) indicate that the type of fertilizer showed a tendency (*p* < 0.1) to influence both the structure and dispersal of the microbial community, while the inoculated bacterium did not present significant effects (*p* = 0.422).

### 2.7. Relative Abundances at Genus Level

As shown in Figure 8, absolute abundance analyses exhibited an increase in the presence of the genera *Bacillus* spp. and *Pseudomonas* spp. in the samples treated with the strains C1 (*Bacillus pretiosus*) and C2 (*Pseudomonas agronomica*), respectively. This increase was consistent regardless of the type of fertilizer applied (water, EDAR or EDARST).

As shown in Figure 9, each dot represents a bacterial genus whose relative abundance differs significantly from the reference group. The direction of change is given on the horizontal axis as the model coefficient, with positive values indicating an increase and negative values a decrease in relative abundance. These results highlight the capacity of certain treatments to modulate the soil microbiome at the taxonomic level.

Differential abundance analysis with ANCOM-BC revealed marked shifts in bacterial composition. Treatment with EDAR promoted enrichment of genera such as *Devosia*, *Planctomicrobium*, *Bifidobacterium*, and other taxa typically associated with nutrient-rich environments. In contrast, EDARST also enriched genera including *Devosia*, *Candidatus Kaiserbacteria*, and TM7, but produced a slight reduction in *Bacillus* spp. When the effect of the inoculated bacteria was assessed, *Pseudomonas agronomica* favored the proliferation of groups such as Fimbrimonadaceae and Pedosphaeraceae, whereas *Bacillus pretiosus* stimulated the growth of taxa including Latescibacterota and Saccharimonadales.

### 2.8. Gene Prediction

Functional prediction using PICRUSt2 revealed potential similar functional profiles between the microbial communities of the different treatments, both in the COG categories and in the KEGG metabolic pathways (Appendix A).

In the results of COG functional categories (Appendix A), the dominant functions were those associated with amino acid metabolism, carbohydrate metabolism, and inorganic ion transport, followed by functions related to DNA repair and replication, energy conversion, and predicted general function. These functions were consistently represented in all the conditions analyzed (EDAR, EDARST, W), with no evident changes in their relative distribution. In the KEGG metabolic classification (Appendix A), genes associated with amino acid metabolism, carbohydrate, cofactor and vitamin metabolism, and energy predominated, along with a significant contribution of secondary metabolite and nucleotide biosynthesis pathways. The relative stability in functional composition suggests that bacterial communities maintain a conserved functional core despite differences between treatments.

A differential analysis of KEGG functions was performed among microbial communities using the non-parametric Kruskal–Wallis test. No function reached significance after correction for multiple tests (*FDR* > 0.05).

A classification analysis was applied using Random Forest on the KO functions predicted by PICRUSt2, with the aim of identifying the most relevant functional genes to discriminate between treatments. The model was trained on 500 trees and tested three predictors per division (mtry = 3), keeping randomness turned on. The overall out-of-bag error (OOB error) was 55.6%, which corresponds to an average accuracy of 44.4%. The confusion matrix indicated that the classification errors were uniform across classes, with no bias toward a specific group. The importance of the variables was evaluated using the Mean Decrease Accuracy statistic, prioritizing the functions with the greatest contribution to the model.

To identify discriminant functions between conditions, a supervised Random Forest model was used, which allowed prioritizing functional genes with greater classificatory power (Appendix A). The most important predicted genes were associated with treatment-specific functions (Appendix A). In the treatment of WATER (irrigation with water, without or with bacteria), regulators of two components of the NarL family were identified, a copper efflux regulator and MFS-type transporters linked to lactate metabolism and ubiquinol oxidase, suggesting a functional response focused on redox maintenance and ionic stress under conditions of low organic load.

In the WATER treatment (irrigation with water, without or with bacteria), regulators of two components of the NarL family were identified, a copper efflux regulator and MFS-type transporters linked to lactate metabolism and ubiquinol oxidase, suggesting a functional response focused on redox maintenance and ionic stress under conditions of low organic load.

In the EDAR treatment (organic residue of meat WWTP, with or without bacteria), the glycan biosynthesis protein was highlighted, while in EDARST (the same sterilized residue, with or without bacteria) a greater diversity of functions was observed, including MFS transporters, oligogalacturonate transport systems, antibiotic resistance, and pathways related to ubiquinone biosynthesis and methanofuran metabolism. These differences point to a specific functional adaptation according to the matrix, with a greater degree of functional specialization in the sterilized waste.

Finally, in order to provide a more detailed representation of microbial functionality, the KEGG modules and the complete KEGG pathways were analyzed (Appendix A).

## 3. Discussion

In the present paper, the effect of the incorporation of a biofertilizer on the growth, nutritional status, and soil microbiota associated with *Q. ilex* seedlings was evaluated. This research was carried out in a natural environment, which enabled the evaluation of the treatment effects in the presence of realistic environmental variability. The current context has fostered an expanding line of research aimed at enhancing the adaptation of forest species, with particular emphasis on plant–microorganism interactions.

In this context, particular attention has been given to plant growth-promoting bacteria (PGPB). Among the soil bacterial genera, *Pseudomonas* spp. and *Bacillus* spp. are widely recognized for their effectiveness in promoting plant growth and improving nutritional status in different species [26,27]. Building on previous laboratory evidence [14], the present study evaluated the field performance of *Bacillus pretiosus* (C1) and *Pseudomonas agronomica* (C2).

In the present study, the selected strains were chosen for their biotechnological relevance, having previously been characterized as plant growth-promoting bacteria (PGPB) [14]. Whole-genome sequencing further confirmed the absence of virulence factors and transmissible antibiotic resistance genes, supporting their safe use in agricultural inoculation [14].

The use of wastewater treatment plant (WWTP) residues in this experiment served a dual purpose: functioning as carriers for PGPB strains and providing a complex nutrient matrix that facilitates nutrient mineralization from organic to inorganic forms, thereby enhancing plant nutrient uptake [28]. Similar strategies have been reported in other studies, where residues such as municipal solid waste, agricultural by-products, and microalgae biomass were reused to reintegrate waste into the productive cycle within a circular economy framework [14,28,29]. The results of the present study revealed a marked improvement in both dry weight and shoot length compared with the controls, particularly under chemical treatment with WWTP residues. Both bacterial strains produced comparable outcomes, with slight variations, notably the greater shoot elongation observed in plants treated with EDARC1. These findings are consistent with previous studies that employed waste sources in combination with bacterial inocula to promote plant growth, such as biofertilizers based on *Pseudomonas* spp. and humic substances from brown coal for pine reforestation [30,31].

Regarding the comparison between WWTP treatments (EDAR and EDARST), which differed in the use of sterilized and non-sterilized matrices, no significant differences were detected among treatments, whether inoculated or not, or in comparison with the controls. This similarity may reflect the soil’s buffering capacity, selectively integrating only microorganisms relevant to the native microbiome, as reported in previous studies [32,33]. Both strains performed comparably under EDAR and EDARST conditions, with a slight but non-significant increase in growth observed in EDARC1. Moreover, a lower plant mortality rate was recorded when the non-sterilized residue was applied, either alone or inoculated with C1 and C2, suggesting that rhizospheric communities and plants benefit more from the residue in its unsterilized state. Overall, the plant development and survival data indicate that sterilization of the residue reduces plant survival, raising the need to evaluate the cost–benefit of sterilization in future biofertilizer applications, given that it entails additional expenses but appears to compromise plant performance.

Although nutritional analysis was not the primary focus of this study, which aimed to assess the potential of biofertilizers in reforestation processes, its inclusion provides a valuable indirect indicator of the overall physiological status of plants. This perspective, still uncommon in agroforestry restoration research, complements the traditional emphasis on growth and biomass variables. In this study, the PCA analysis (Figure 3) revealed notable trends: clear segregation patterns were observed between treatments, both by irrigation type and by the application of beneficial bacteria, indicating functional changes in plant physiology. While some degree of overlap is expected due to natural soil and environmental variability, the consistent separation of PGPB treatments from the control highlights a measurable influence on the nutritional profile.

When evaluating biofertilizers, a central consideration is whether inoculation may disrupt resident soil microbial communities [34]. The functional diversity profiles obtained with Biolog EcoPlates (Figure 4) indicate that the PGPB strains tested did not negatively affect the metabolic performance of the soil microbiota. Rather than displacing native groups, the inoculated bacteria appear to coexist with them while preserving the overall balance of the system. Similar outcomes have been reported in wheat, where inoculation with a *Pseudomonas* sp.–*Azotobacter* sp. consortium left community diversity essentially unaltered [35], and in oilseed rape, where *Pseudomonas* sp. increased the abundance of beneficial groups without reducing overall diversity [36]. In the present study, diversity indices remained within comparable ranges across treatments, suggesting that the introduction of biofertilizers does not compromise microbiome stability. This observation is noteworthy, as it supports the preservation of established metabolic interactions and the ecological balance of the soil. Nevertheless, contrasting results have been reported: Sierra-García [37] and Ferreira [38] documented alterations in functional and metabolic diversity following the introduction of exogenous microbial strains.

In addition, the functional profile grouped by nutrient families (Figure 5) reveals more efficient patterns of use, highlighting a higher consumption of carbohydrates and carboxylic acids in treatments with WWTPs, possibly associated with an intensification of organic matter mineralization. Carbohydrate consumption may indicate an intensification of the use of easily assimilated carbon sources [39,40]. Reduced amino acid consumption in EDAR and EDARST treatments could suggest an improvement in mineral nitrogen availability, reducing the need for microbial communities to metabolize nitrogenous organic compounds [41]. Furthermore, the higher consumption of carboxylic acids observed under inoculation with *Bacillus* (C1) in WWTPs suggests an intensification in the mineralization of compounds related to the carbon cycle, which is an acceleration of the transformation of organic matter into CO_2_ and CH_4_, which other authors have observed as well with the use of *Bacillus mucilaginosus* [42]. Overall, the exhibited functional profile shows no signs of disruption caused by the inoculation of PGPB strains but reflects a more efficient and consistent pattern of resource use with functional improvement of rhizospheric microbial communities.

Another aspect of particular relevance in this study is the analysis of antibiotic sensitivity, given the growing scientific and societal concern regarding the intensive use of these compounds in agricultural settings [43]. The accumulation of antibiotics and the consequent selection of resistant bacteria represent an emerging threat to public health, as soil constitutes a major reservoir of resistance genes [43]. It is therefore essential to include such assessments in studies of this nature to confirm that biofertilizer application does not exacerbate the prevalence of resistance genes in treated soils [44].

In this study, the cenoantibiogram was employed as a technique to evaluate antibiotic sensitivity, providing more ecologically meaningful outcomes since it assesses the resistance of the entire microbial community [45]. This contrasts with other approaches such as disc diffusion or microdilution to estimate MIC values, as well as genotypic methods (PCR, microarrays, sequencing) for the detection of resistance genes [46]. The data obtained demonstrated that both fertilizer type and bacterial inoculation influence community resistance to antibiotics. Moreover, the incorporation of beneficial strains (C1 and C2) reduced the resistance profile, particularly when applied in complex matrices such as EDAR and EDARST, probably due to the absence of microbial competition. These findings highlight the potential application of beneficial bacteria as a strategy for mitigating antimicrobial resistance in agricultural and forestry environments [47]. Comparable results have been reported in studies with *Peribacillus frigoritolerans* subsp. *mercuritolerans* [48], *Pseudomonas mercuritolerans* [49], and consortia of *Pseudomonas* strains [45], where the addition of isolates with low MIC profiles mitigated the resistance of microbial communities exposed to mercury.

A metagenomic study of amplicons was conducted through sequencing of the 16S rRNA gene. This approach enabled the characterization of microbial communities in terms of both identification and quantification of microbial taxa. It also revealed community patterns that could remain undetected by traditional methods, thereby providing more comprehensive and ecologically relevant data [47]. Within this study, metagenomics served as a key tool for interpreting diversity and abundance, establishing a solid basis for assessing the ecological impact of the applied treatments.

The Shannon index of taxonomic diversity (α diversity) showed no significant differences between treatments, indicating that the richness and balance of bacterial communities were not compromised by the application of fertilizers or the inoculation of strains. This outcome further supports the feasibility of integrating PGPBs without altering soil biodiversity. Some studies corroborate this possibility [46], whereas others report that the introduction of exogenous agents can modify the diversity of the environments where they are introduced [50]. These findings underscore the importance of applying diversity analyses, as they demonstrate the capacity of the two strains tested to adapt to their environment and exert beneficial effects without altering its composition. Regarding β diversity, PCoA analysis revealed partial separation between groups according to fertilizer type, although this trend was not statistically significant. Similarly, PERMANOVA confirmed the absence of significant differences between treatments, suggesting that fertilizer type did not markedly affect the structure of microbial communities.

At the genus level, relative abundance analysis showed a clear and consistent increase in *Bacillus* spp. and *Pseudomonas* spp. in samples treated with the C1 and C2 strains, respectively. This result confirms the colonization efficiency of the inoculated strains, irrespective of fertilizer type, consistent with previous findings demonstrating that specific strains can be effectively incorporated into soil microbial communities without causing major disruptions to the native microbiota [51,52]. The integration of microbial species into soil communities as biofertilizing agents is particularly valuable when they do not displace other dominant genera, as this supports microbial stability and diversity, which are essential for microecosystem health [35].

In line with these results, the application of selected strains appeared to favor the enrichment of beneficial microbial groups without disrupting overall bacterial balance. Specifically, *Pseudomonas agronomica* promoted an increase in genera such as *Fimbrimonadaceae* and *Pedosphaeraceae*, whereas *Bacillus pretiosus* was associated with the stimulation of groups including *Latescibacterota* and *Saccharimonadales*. This points to a strain-specific modulatory influence on soil microbial communities, consistent with previous reports indicating that different plant growth-promoting bacteria can reshape soil microbiome structure and function in distinctive ways [53].

Conversely, treatment with EDARST resulted in the enrichment of taxa such as *Devosia*, *Candidatus Kaiserbacteria* and *TM7*, but also in a slight decline in *Bacillus* spp. This effect may be attributable to the elimination of the biologically active fraction of the fertilizer through sterilization, which could reduce the abundance of beneficial microorganisms involved in plant protection and pathogen competition [54].

The overall stability observed in functional profiles across treatments suggests the existence of a resilient microbial functional core, a common feature of soil microbiota that retains conserved metabolic functions. Nonetheless, supervised analyses revealed distinct functional signals indicative of adaptive responses to the physicochemical properties of each matrix.

In WC0, WC1, and WC2 treatments, discriminant functions were primarily associated with two-component regulatory systems and electron transport, reflecting microbial communities adapted to relatively simple environmental stimuli where redox gradients and oxygen availability act as principal modulators. The detection of MFS-type transporters and ubiquinol oxidase suggests a metabolism oriented toward homeostatic maintenance and energy efficiency under nutrient-poor conditions.

By contrast, EDAR treatments were enriched in structural biosynthesis pathways (e.g., glucans), consistent with adaptation to organic-dense matrices characterized by higher osmotic pressure and complex carbon availability. These conditions promote functions associated with extracellular matrix modification and biofilm formation.

The most functionally diverse scenario was detected in sterile waste (EDARST). Activation of pathways linked to oligosaccharide transport, antibiotic resistance, and ubiquinone biosynthesis likely reflects the combined effects of residual carbon availability, absence of native microbial competition following sterilization, and inoculation with specific strains. This condition favors communities that exploit secondary metabolic pathways and actively compete for ecological niches through exclusion mechanisms and fine-scale adaptive strategies.

Overall, the results demonstrate that although the core functional capacity of bacterial communities remains conserved, accessory and adaptive functions are strongly shaped by both matrix composition and colonization history (sterilization and reinoculation). These outcomes carry direct implications for biostimulation and bioremediation strategies, highlighting the role of matrix selection in steering not only microbial composition but also the functional traits expressed by the soil microbiome.

## 4. Materials and Methods

### 4.1. Bacterial Strains

In this study, two strains were tested: *Bacillus pretiosus* (C1) and *Pseudomonas agronomica* (C2). These strains were described and characterized as PGPB (Table 3) by the “Environmental Microbial Biotechnology” (MICROAMB) research group of the Faculty of Pharmacy of the CEU San Pablo University [14]. Whole-genome sequencing conducted in earlier studies [14] confirmed the absence of virulence factors and genes associated with transmissible antibiotic resistance.

### 4.2. Experimental Design: Irrigation Matrices (Chemical Treatment) and Biological Treatment (Strains)

The field trial was conducted at the CEU Montepríncipe research site (40.403826, −3.836913), using one-year-old *Quercus ilex* seedlings provided by the Madrid Institute for Rural, Agrarian and Food Research and Development (IMIDRA). A total of 35 plants were established per treatment. Three irrigation matrices were evaluated: water (W), organic waste from a wastewater treatment plant (EDAR), and sterilized EDAR (EDARST). Each matrix was combined with three biological treatments: control without inoculum (C0), inoculation with *Bacillus pretiosus* (C1), and inoculation with *Pseudomonas agronomica* (C2). Treatments were applied monthly by irrigation, while routine drip irrigation was maintained to ensure seedling survival.

### 4.3. Preparation of Bacterial Suspensions and Biofertilizers

#### 4.3.1. Preparation of Bacterial Suspensions

Starting from pure cultures of the C1 (*Bacillus pretiosus*) and C2 (*Pseudomonas agronomica*) strains in nutritional agar from Condalab^®^ (Madrid, Spain), a bacterial culture was prepared in LB liquid medium. After 48 h of growth, the bacterial density was checked at 0.5 on the McFarland scale (10^8^ cfu mL^−1^) using UricultTM submersible paddles (Liofilchem srl, Roseto degli Abruzzi, Italy). This process ensured the standardization of the bacterial inoculum for subsequent application in the final volume of the irrigation matrix.

#### 4.3.2. Preparation of Biofertilizers

The WWTP waste (Industrias Cárnicas Villar, S.A., Soria, Spain) was used at a concentration of 1/512 volume/volume (V_WWTP_/V_H2O_). The physicochemical composition of the organic waste can be consulted in Table 4. For the preparation of the sterilized organic waste (EDARST), to eliminate the microbiota of the residue, it was sterilized by autoclaving (121 °C, 20 min, 1 atm). The biofertilizer was prepared monthly to prevent bacterial growth and transformations. For the synthesis of the biofertilizer, 100 mL of a 0.5 bacterial McFarland suspension of each strain, *Bacillus pretiosus* (C1) or *Pseudomonas agronomica* (C2) was added, except for the irrigation matrix that only had 900 mL of H_2_O. Next, 20 mL/L of the organic waste was added.

#### 4.3.3. Plant Growth Conditions and Irrigation Regime

The trial was conducted in the field with a duration of 1 sap (12 months). Each plant was irrigated with 500 mL of biofertilizer which includes chemical treatment (EDAR or EDARST) and biological treatment (*Bacillus pretiosus* or *Pseudomonas agronomica*). Monthly watering was carried out with the biofertilizer, and drip irrigation was used as maintenance irrigation (from April 2024 to April 2025 harvest).

#### 4.3.4. Harvesting, Total Biomass Measurement and Nutritional Analysis

This process involved the destructive extraction of the aerial and radical part of each plant. From the root fraction, a sample of rhizospheric soil (2 g per seedling) was obtained for subsequent analysis. To determine total biomass, the harvested plants were left to dry at room temperature (22 °C ± 2 °C) for one week, recording the dry weight in grams (g). The size of the plants was measured in situ, in the same way the number of surviving plants at the time of harvest was counted. For nutritional analysis, the leaves were separated from the stem manually. Each fraction was divided into three replicates of the complete plant (replications). The samples were packaged and kept refrigerated for shipment to the Rock River Labs Spain analysis laboratory, located in Lalín, Pontevedra, Spain. The parameters measured in the nutritional analysis were proteins (%DM), total amino acids (%BW), minerals (%DM), carbohydrate digestibility (%DM), sugars (%DM), fiber digestibility (%DM), and fatty acids (% Total FA).

### 4.4. Extraction of Soil Microbial Communities

For the extraction of the rhizospheric communities, the procedure developed by García-Villaraco (2009) was followed [55]. To this end, 2 g of rhizospheric soil was suspended in 20 mL of sterile saline solution (0.45% NaCl) and homogenized with an Omni-Mixer homogenizer at 16,000 r.p.m. for 2 minutes. It was then centrifuged at 690× *g* for 5 minutes with a Hettich Zentrifugen centrifuge model Mikro 22R (Andreas Hettich GmbH & Co, Tuttlingen, Germany).

### 4.5. Antibiotic Susceptibility Test—Cenoantibiogram

An analysis of the cenoantibiogram was performed, which is a study of the antibiotic susceptibility of the entire microbial community [45]. A soil extract was made in saline solution (NaCl 0.45%) with a density of viable microorganisms greater than 10^8^ ufc.mL^−1^, with an optical density (OD) of 0.5 on the McFarland scale. Next, the seeding was carried out on Mueller–Hinton agar (Condalab^®^, Madrid, Spain) and the minimum inhibitory concentration (MIC) was evaluated using antibiotic strips from ε test for several antibiotics: amoxicillin-clavulanic acid (AUG), piperacillin-tazobactam (TZP), imipenem (IMI), imipenem-EDTA (IMD), ciprofloxacin (CIP), cefotaxime (CTX) and gentamicin (CN) (BioMérieux^®^, Marcy l’Etoile, France). The plates were incubated according to the manufacturer’s instructions, at 25 °C, and the minimum inhibitory concentration (MIC) was quantified using the most restrictive inhibition halo as a reference.

### 4.6. Study of the Functional Diversity of the Microbial Community

From the soil extract, the Biolog Eco^®^ (Biolog, Inc., Hayward, CA, USA) plates were inoculated using 135 μL per well. The plates were incubated for 168 h at a temperature of 25 °C ± 2 °C, measuring absorbance at 492 nm every 24 h using the Asys UVM340 plate reader (Benchmark Scientific Inc., NJ, USA) in conjunction with Micro WinTM V3.5 software. The AWCD (Average Well Color Development) value [56] was plotted as a function of the incubation time, which allowed us to obtain the growth curves of the microbial communities present in the plates. From these curves, the incubation time corresponding to the beginning of the stationary phase of microbial growth (120 h) was selected. In addition, using the values of corrected absorbance in that selected time (AWDC), the metabolic diversity of each sample was calculated using the Shannon–Weaver diversity index [57].$$H\left(m\right)=-\sum q_{i}\log_{2}q_{i}$$

Knowing that *qi* = *n/N*, “*n*” is the corrected absorbance (AWCD) and *N* is the total absorbance of all wells.

### 4.7. Nutritional and Biometric Analysis

To assess the variation in biomass dry weight according to different types of irrigation, a one-factor ANOVA test was applied. Similarly, a principal component analysis (PCA) of the nutritional variables was carried out. All analyses were performed using SPSS v.30.0 software (IBM Corp, Armonk, NY, USA).

### 4.8. Metagenomics: Bioinformatic Processing and Statistical Analysis

#### 4.8.1. Metagenomics

The composition and structure of the sampled microbial communities was evaluated by amplification and sequencing of the variable regions V3-V4 of the *16S rRNA* gene. Amplification was performed after 25 cycles of PCR. In this procedure, positive (CM) and negative (NC) controls were used to ensure quality control. The positive control is a mock community and was processed in the same way as the samples. The libraries obtained were sequenced using Illumina Miseq (300 × 2).

#### 4.8.2. Bioinformatics Processing and Statistical Analysis

A schematic view of the process can be found in Figure 10. Demultiplexed crude readings, both forward and reverse, were processed using QIIME2 (v2025.4) [58].

Beta diversity distance matrices were used to represent the PCA. The significance of the groups was tested using the Permanova and ANOSIM tests. The Permdisp test was used to identify location vs. scattering effects [59]. The significance threshold (*p*-value) was set at 0.05. The taxonomic assignment of the phylotypes was performed using a Bayesian Classifier [60] trained with Silva version 138 database (full-length sequences at 99% OTUs) [61]. The differential abundance of taxa was tested using Generalized Linear Models with Negative Binomial Distribution. A Generalized Linear Model was constructed using the R MASS v.7.3-54 package [62]. The significance threshold (*p*-value) was set at 0.05.

#### 4.8.3. Gene Prediction

A functional prediction analysis of genes was performed from sequencing data using PICRUSt2 (Phylogenetic Investigation of Communities by Reconstruction of Unobserved States, version 2). The processing and analysis of the data were carried out in Python 3, using the scripts and modules specific to the execution of PICRUSt2, in order to predict the functional abundance of genes in the analyzed samples.

Before performing the non-parametric Kruskal–Wallis test, the ARiSTa (Adaptive Rank-based Inverse Score Transformation analysis) was applied, to normalize the functional abundances and guarantee the statistical validity of the analysis. After the transformation, a Kruskal–Wallis test was conducted to identify differentially abundant functions between the experimental groups.

Finally, a classification model based on Random Forest was implemented to identify the most important functional variables that discriminate between groups.

## 5. Conclusions

The use of WWTP residue (sterilized and unsterilized) suggested a positive effect on holm oak (*Quercus ilex*) growth, as reflected in the increases in size and dry weight compared with irrigation with water alone. Similarly, inoculation with the PGPB strains *Bacillus pretiosus* (C1) and *Pseudomonas agronomica* (C2) appeared to enhance plant development relative to controls without inoculation. The combined application of WWTP residue with both strains showed comparable potential for future use in promoting *Q. ilex* growth.

Moreover, the biofertilizer derived from the combination of PGPB strains and organic residue appeared to support plant establishment and adaptation to the soil. Data on seedling survival further suggest that non-sterilized WWTP residue favored higher survival rates compared with its sterilized counterpart, indicating that the biologically active fraction of the biofertilizer may play a role in treatment success. As sterilization involves additional costs and appears to reduce survival, its necessity should be reconsidered in future applications. Nevertheless, long-term field trials with larger sample sizes are required to validate these findings.

Cenoantibiogram analyses indicated a reduction in soil resistance profiles following inoculation with *B. pretiosus* and *P. agronomica*, both individually and in combination with EDAR/EDARST residues. In particular, the EDARC1 treatment appeared especially effective, suggesting potential as a biofertilizer for agroforestry recovery. At the same time, the ecological compatibility of the approach was supported, since application of the biofertilizer did not markedly alter the balance of the native microbial community. Soil diversity and metabolic functions remained stable, while metagenomic analyses indicated persistence and integration of the inoculated strains within the holm oak rhizosphere. The α-diversity patterns suggested that the edaphic community accommodated the introduced strains without major disruption, pointing to the potential of the selected bacteria to coexist in a complex, non-sterile microbiome.

## Figures and Tables

**Figure 1 plants-14-02654-f001:**
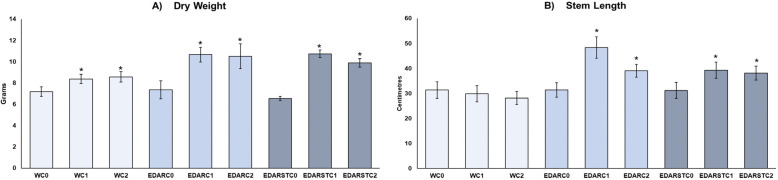
Biometric analysis of (**A**) dry weight of the stem (g) and (**B**) length of the stem (cm). The colors indicate the type of irrigation: EDAR: irrigation with unsterilized waste. EDARST: irrigation with sterilized waste. While the intensities distinguish the biotreatment: C1: *Bacillus pretiosus*. C2: *Pseudomonas agronomica*. C0: irrigation with water. The error bars indicate the adjusted standard deviation for ease of reading. * Represents significant differences (*p* < 0.05) when compared to intratreatment controls.

**Figure 4 plants-14-02654-f004:**
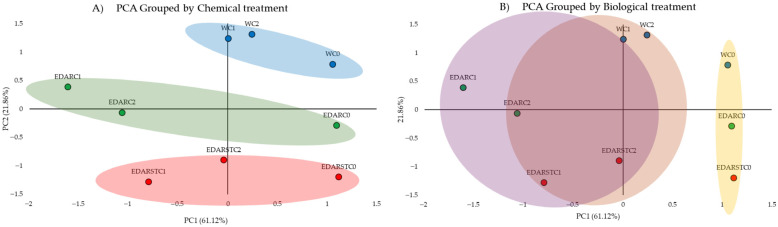
This principal component analysis explores the multivariate structuring of medium resistance to antibiotic groups (fluoroquinolones, beta-lactams, aminoglycosides, carbapenems) under different combinations of chemical (TQ) and biological (TB).

**Figure 5 plants-14-02654-f005:**
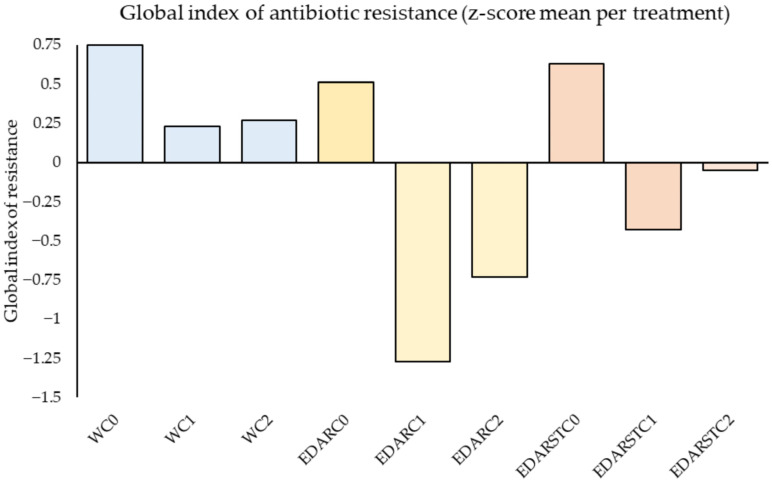
Summary of the values of the mean community resistance of the soil against different families of antibiotics, standardized by z-score for each treatment. Positive values indicate greater resistance compared to the general average, while negative values reflect a cleaner profile or a lower resistance load. The colors represent the chemical irrigation treatment: blue for WATER, yellow for EDAR, and orange for EDARST.

**Figure 6 plants-14-02654-f006:**
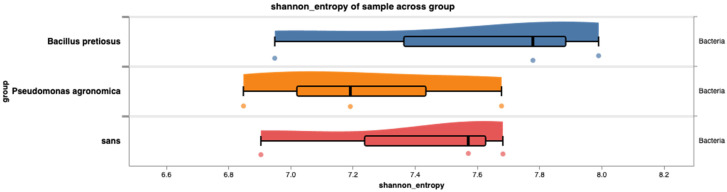
Raincloud plots show the distribution of the subjects’ Shannon entropy measurement in the different groups. Estimation of core density made using a bandwidth calculated by Scott’s method. Box charts show the minimum and maximum of the data (whiskers), as well as the first, second (median), and third quartile (box).

**Figure 7 plants-14-02654-f007:**
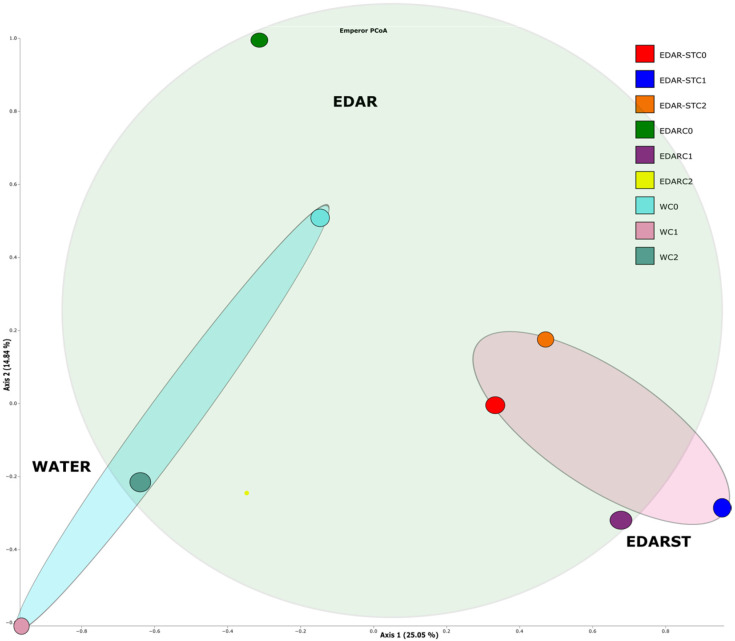
Principal coordinate analysis (PCoA) based on weighted distances from UniFrac to evaluate differences in microbial community structure according to fertilizer type and inoculated bacteria.

**Figure 8 plants-14-02654-f008:**
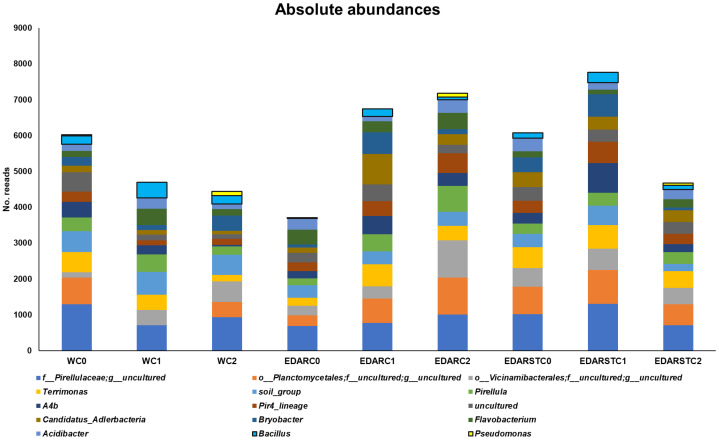
Absolute abundance of the 13 most abundant microbial genera and the genera of the inoculated strains. The data correspond to the total number of readings assigned by genus, normalized by sample. The most representative genera are shown, with *Pseudomonas* spp. in yellow and *Bacillus* spp. in blue.

**Figure 9 plants-14-02654-f009:**
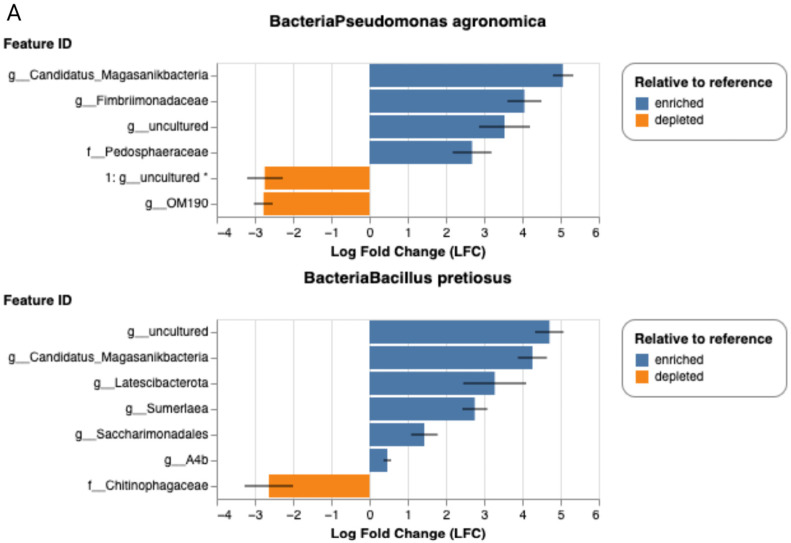
Analysis of differential abundance of bacterial genera by ANCOM-DC according to the treatment applied. The significantly enriched (in blue) or depleted (in orange) taxa (*q* < 0.05) are shown according to (**A**) the inoculated bacteria (*Bacillus pretiosus, Pseudomonas agronomic* and control without inoculation) and (**B**) the type of fertilizer applied (water, EDAR, EDARST).

**Figure 10 plants-14-02654-f010:**
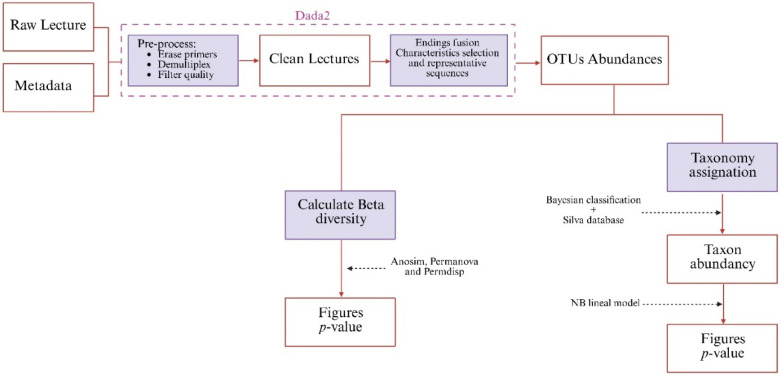
Sequence processing and analysis sequence.

**Table 1 plants-14-02654-t001:** Mortality percentage after 1 year.

	WC0	WC1	WC2	EDARC0	EDARC1	EDARC2	EDARSTC0	EDARSTC1	EDARSTC2
Planted	35	35	35	35	35	35	35	35	35
Survival after 1 year	21	18	21	22	22	24	23	19	13
Mortality rate	40%	49%	40%	37%	37%	31%	34%	46%	63%

**Table 2 plants-14-02654-t002:** Mann–Whitney U between groups ((see facet) vs. (see facet)), with automatic bilateral *p*-value calculations and correction for multiple comparisons (*q*-value).

Facet	Group A (See Facet)	Group B (See Facet)	A (Median)	B (Median)	Test Statistic	*p*-Value	*q*-Value
Bacteria	C1	C2	7.77	7.19	7	0.4	0.80
Bacteria	C1	C0	7.77	7.57	7	0.4	0.80
Bacteria	C2	C0	7.19	7.57	3	0.7	0.84
Fertilizer	EDAR	EDARST	7.67	7.57	4	1.00	1.00
Fertilizer	EDAR	Water	7.67	6.95	6	0.70	0.84
Fertilizer	EDARST	Water	7.57	6.95	7	0.40	0.80

**Table 3 plants-14-02654-t003:** PGPB characteristics of the strains. IAA: production of 3-indoleacetic acid; ACCd: production of 1-aminocyclopropane-1-carboxylate deaminase; p/a: presence (+)/absence (−). WGS: Whole-genome sequence (Illumina^®^, Illumina Inc., San Diego, CA, USA).

Identification (WGS)	IAA (µg.mL)	ACCd (p/a)	Siderophores(p/a)
*Bacillus pretiosus*	5.61 ± 0.03	−	+
*Pseudomonas agronomica*	5.85 ± 0.09	+	+

**Table 4 plants-14-02654-t004:** Physicochemical composition of the WWTP waste. (Analysis carried out by LabAqua. Tests covered by accreditation ENAC nº109/LE 28.).

	Parameters	Methods	Results	Units
Physicochemical characteristics	Conductivity at 20 °C	A-F-PE-0015 Electrometry	1454	µS/cm
	Conductivity at 25 °C	A-F-PE-0015 Electrometry	1612	µS/cm
	Biochemical Oxygen Demand (BOD5)	A-F-PE-0002 Gauge method	3200	mgO_2_/L
	Chemical Oxygen Demand	A-F-PE-0003 Digestion—Colorimetry	6720	mgO_2_/L
	Chemical demand for decanted oxygen	A-F-PE-0003 Digestion—Colorimetry	4220	mg/L
	Nitrates	A-F-PE-0010 Digestion	<0.05	mg/L
	Kjeldhal Nitrogen	A-F-PE-0007 Kjeldhal	296.7	mg/L
	pH	A-F-PE-0010 Electrometry	6.5	U.pH.
	Suspended solids	A-F-PE-0006 Gravimetry	3228	mg/L
	Toxicity	PIT-F/0012 Bioluminescence assay with *Vibrio fisheri*	14	U.T.
Majority cations	Potassium	A-D-PE-0025-ICP-OES	60.2	mg/L
Anions	Nitrates	A-BV-PE-0001 HPLC—Conductivity	<2.5	mg/L
	Orthophosphates	Ca-R-PE-0011 Spectrometry	74.32	mg PO_4_/L
	Sulfates	A-BV-PE-0001 HPLC—Conductivity	89.0	mg/L
	Sulfites	A-F-PE-0040 Volumetry	4.5	mg/L

## Data Availability

The metagenomic sequences are deposited in the NCBI repository under de Bioproject number. The accession numbers for the strains used are JAOSHO01 (*Pseudomonas agronomica*) and JAOXJG01 (*Bacillus pretiosus*).

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
