# Peer review of "Biotechnological Test of Plant Growth-Promoting Bacteria Strains for Synthesis of Valorized Wastewater as Biofertilizer for Silvicultural Production of Holm Oak (Quercus ilex L.)"

_plants, 2025, doi:10.3390/plants14172654_

Round 1
Reviewer 1 Report
Comments and Suggestions for Authors
This manuscript explores the impact of two biofertilizers derived from wastewater treatment enriched with two PGPB on the growth of oak plants under field conditions. The study also determined soil microbes and antibiotic resistance in the soil. The study addresses a relevant topic in the context of restoring degraded forest soils. Although it is well written and presents important findings, the manuscript requires revision to improve clarity, rigor, and quality. One of the problems reported above depends on the position of the Materials & Methods section (the numbering of figures and tables, as well as the use of abbreviations).
Other comments
Line 19: The scientific name of the species should be italicized. Revise the text accordingly.
Line 31: I think you do not need to repeat some of the words from in the title.
Line 48: Uniforme the presentation of two references in the text. See lines 48 ([5-6]) and line 64 ([14,15]], for example.
Line 127: It should read "Figure 1." Revise the numbering of figures throughout the text.
Line 129: “C0” should be included in the the header of the figure. The YY axis should include the units.
Line 132: How much higher than the control? Include the percentage increase compared to the control.
Line 141: It should read "Table 1." The table should appear after the text. Replace the term "marras" with a more general term.
Line 142: Revise the "rate of growth." It does not represent the content of the table.
Line 143: The expression “decreases significantly” is not evident in the results. A statistical analysis should be presented. The entire sentence should be revised.
Line 161: The figure header should be more detailed. See lines 571–574 for uniformization.
Line 170 and Line 177: The meanings of “TQ” and “TB” should be explained the first time they appear.
Line 198: The standard deviation should be included.
Line 209: The figure should be more legible. Please see comment on line 148.
Line 212: Separate the headers of Figure 7 and Table 5. When referring to water, should it be "C0"? What do “test-statistic” and “q-value” mean?
Line 215-218: The results of “q-value” and "test statistic" should be referenced in the text.
Line 223: Revise the reference to the Figure.
Line 229: What does "NaN" mean in the table?
Line 240: Improve the resolution of the figure. Include the units.
Line 250: Improve the resolution and organization of the figure.
Line 260-266: Throughout the texto, different terms are used to express the same thing. Revise to be consistent.
Line 277: Revise the figure number. The figure should be formatted. It is illegible.
Line 325-328: Desnecessary.
Line 330: “seedlings was evaluated” is not in italics.
Line 335-344: This is explained in the Material & Methods section. Not necessary.
Line 350: The percentage of increment should be provided.
Line 352: This should be reflected by a statistical analysis that comparing both PGPB species.
Line 355: Should it be “WWTP” or “EDAR”?
Line 361-363: Present a statistical analysis that compares both treatments. In Figure 2, you only compare with WC0.
Line 367: The meaning of “rate of treatment” should be included or reworded.
Line 450: It should be “[45]”.
Line 459: The “other studies” referred to should be presented in the text.
Line 467-470: The sentence should be in English.
Line 480: Consider removing “WWTP”.
Line 490: “Water treatment” should be referred as “WC0, WC1 and WC2”.
Line 496: Consider removing “WWTP residue”.
Line 519: It should be “[14]”.
Line 523: Uniformize “p/A” and “p/a”.
Line 529: Clarify the meaning of “sap”.
Line 545: Clarify the meaning of “1/512 (VEDAR/vH2O)”.
Line 548: “EDAR_ST” should be “EDAR”.
Line 540, 549, and 595: Uniformize the presentation of temperature units.
Line 576: It should read “Garcia-Villaraco [54]”.
Line 580: Include the manufacturer of the equipment. Revise the text accordingly.
Line 642: The conclusion section needs some refinement to better reflect the findings of the study (see the following comments). Additionally, the interpretation of the results should go further. For example, it would be important to clearly indicate which PGPB strain demonstrated superior behaviour, and to indicate the pratical implications of omitting the sterilization process and strategies to solve it. In addition, it would be important to emphasize the need for long-term field trials to validate the results.
Line 648-654: Rewrite the entire paragraph to take into account the text (lines 132–140) and Figure 2 (Line 127).
Line 670-674: Rewrite the sentence because the table reports the percentage of survival without statistical analysis.
Line 701: The reference list should be revised to meet the journal guidelines. Several references are not formatted correctly.
Author Response
Line 19: The scientific name of the species should be italicized. Revise the text accordingly.
Done
Line 31: I think you do not need to repeat some of the words from in the title.
Thank for the appreciation bus Biofertilizer, WWTP and PGPB are part of the essential core of the manuscript and we consider it important as keywords.
Line 48: Uniforme the presentation of two references in the text. See lines 48 ([5-6]) and line 64 ([14,15]], for example.
Changed and revised through the manuscript.
Line 127: It should read "Figure 1." Revise the numbering of figures throughout the text.
Changed and revised through the manuscript.
Line 129: “C0” should be included in the the header of the figure. The YY axis should include the units.
Corrected.
Line 132: How much higher than the control? Include the percentage increase compared to the control.
Information added: “For dry weight it was found an increment respect the control with water (WC0) with WC1, WC2, EDARC1, EDARC2, EDARSTC1 and EDARSTC2 of 16.5%, 19.2%, 48.4%, 46.2%, 49.4% and 37.4% respectively. And for stem lengths of WC0 with EDARC1, EDARC2, EDARSTC1 and EDARSTC2 of 54.2%, 24.7%, 25.3% and 21.8% respectively.”
Line 141: It should read "Table 1." The table should appear after the text. Replace the term "marras" with a more general term.
Changed.
Line 142: Revise the "rate of growth." It does not represent the content of the table.
Changed for rate of deaths.
Line 143: The expression “decreases significantly” is not evident in the results. A statistical analysis should be presented. The entire sentence should be revised.
Sentence rewrite to: “…decreases when EDAR treatments are used…”
Line 161: The figure header should be more detailed. See lines 571–574 for uniformization.
Changed.
Line 170 and Line 177: The meanings of “TQ” and “TB” should be explained the first time they appear.
Done
Line 198: The standard deviation should be included.
Z-score doesn’t have standard deviation.
Line 209: The figure should be more legible. Please see comment on line 148.
The figure meets the standards set by the journal for its online publication. We understand that in the word version it may seem small, but the image has 600dpi resolution.
Line 212: Separate the headers of Figure 7 and Table 5. Done
When referring to water, should it be "C0"?
When referring to water, it is referred to the chemical matrix and when we name it as C0 it is referred to the biological treatment. In the case of this study its nearly the same. To maintain the nomenclature and a better understanding we prefer to refer the absolute control (water without inoculum) as WC0, because we have the internal control for each chemical treatment (EDARC0 and EDARSTCO) and the water treatments with bacteria (WC1 and WC2).
What do “test-statistic” and “q-value” mean? Line 215-218: The results of “q-value” and "test statistic" should be referenced in the text.
Test-statistic is the statistic of the Mann–Whitney U test (also called Wilcoxon rank-sum). It represents the value calculated from the sums of the ranges between the two groups you are comparing. This number, by itself, does not indicate significance; It serves as the basis for calculating the p-value. If you were to do it manually, it is obtained from the ranges assigned to all the combined values and then adding those of each group to evaluate whether the distributions differ. -value is the p-value adjusted for False Discovery Rate (FDR) control. QIIME2 applies corrections for multiple comparisons (usually the Benjamini–Hochberg method), so that the q-value reflects the probability that a significant result is a false positive, given the total number of tests performed.
In short, Test.statistic is the number that gives you the magnitude and direction of the effect according to Mann–Whitney. And q-value is the p-value corrected for multiple comparisons; It's the one you should use to decide if the difference is statistically significant.
Line 223: Revise the reference to the Figure.
Done.
Line 229: What does "NaN" mean in the table?
Table change for a better understanding of the data.
Line 240: Improve the resolution of the figure. Include the units.
Done.
Line 250: Improve the resolution and organization of the figure.
Done
Line 260-266: Throughout the text, different terms are used to express the same thing. Revise to be consistent.
Revised.
Line 277: Revise the figure number. The figure should be formatted. It is illegible.
Done and move to supplementary material
Line 325-328: Desnecessary.
Deleted.
Line 330: “seedlings was evaluated” is not in italics.
Changed.
Line 335-344: This is explained in the Material & Methods section. Not necessary.
We consider that part as a good introduction of the discussion section because, following the journal organization, the M&M section comes at the end of the manuscript and it gives context to the following ideas.
Line 350: The percentage of increment should be provided.
Information added in results section
Line 352: This should be reflected by a statistical analysis that comparing both PGPB species.
There is no statistical differences between EDAR+inoculum treatments bus it can observed a visual slight difference, as is indicated in the manuscript.
Line 355: Should it be “WWTP” or “EDAR”?
Corrected to EDAR
Line 361-363: Present a statistical analysis that compares both treatments. In Figure 2, you only compare with WC0.
In figure 2 (actual Figure 1) it can be observed a slight growth but in the statistical analysis it doesn’t have sufficient statistical differences between the other treatments. We only want to remark that visual difference and as we indicate in the manuscript, not possess by itself sufficient statistical differences with the other treatments with EDAR or EDARST and the inoculum.
Line 367: The meaning of “rate of treatment” should be included or reworded.
Changed to rate of deaths
Line 450: It should be “[45]”.
Changed
Line 459: The “other studies” referred to should be presented in the text.
Deleted
Line 467-470: The sentence should be in English.
Changed
Line 480: Consider removing “WWTP”.
Done
Line 490: “Water treatment” should be referred as “WC0, WC1 and WC2”.
Done
Line 496: Consider removing “WWTP residue”.
Done
Line 519: It should be “[14]”.
Changed
Line 523: Uniformize “p/A” and “p/a”.
Done
Line 529: Clarify the meaning of “sap”.
Changed to “year”
Line 545: Clarify the meaning of “1/512 (VEDAR/vH2O)”.
Done
Line 548: “EDAR_ST” should be “EDAR”.
Changed
Line 540, 549, and 595: Uniformize the presentation of temperature units.
Done
Line 576: It should read “Garcia-Villaraco [54]”.
Changed
Line 580: Include the manufacturer of the equipment. Revise the text accordingly.
Added
Line 642: The conclusion section needs some refinement to better reflect the findings of the study (see the following comments). Additionally, the interpretation of the results should go further. For example, it would be important to clearly indicate which PGPB strain demonstrated superior behaviour, and to indicate the pratical implications of omitting the sterilization process and strategies to solve it. In addition, it would be important to emphasize the need for long-term field trials to validate the results.
Done.
Line 648-654: Rewrite the entire paragraph to take into account the text (lines 132–140) and Figure 2 (Line 127).
Done.
Line 670-674: Rewrite the sentence because the table reports the percentage of survival without statistical analysis.
Done.
Line 701: The reference list should be revised to meet the journal guidelines. Several references are not formatted correctly.
Revised and cheked
Reviewer 2 Report
Comments and Suggestions for Authors
The article focuses on “biotechnological test of a PGPB strain for the synthesis of a valorised wastewater as biofertilizer for the silvicultural production of holm oak (Quercus ilex L.)”. This type of knowledge is emphasized due to the growing interest in bacteria that can support various sectors of the economy. Many research works focus on the mechanisms of action of bacterial strains with different biostimulating mechanisms. As the authors of this paper emphasize “..it is crucial to use sustainable strategies to restore soils and promote plant growth, thus supporting ecosystem regeneration”. The microbial communities inhabiting the soil, also known as the soil microbiome, have a significant impact on plant physiology and health. Plants derive their nourishment from the soil, but nutrients are available to them thanks to a vast diversity of microorganisms. Bacteria that can promote plant growth include both free-living species and endophytic bacteria that colonize parts of plant tissues. Interactions between soil bacteria and plants and their influence on soil chemical and physical properties are crucial for soil ecosystems.
The authors rightly draw attention to the fact that “…the main challenges faced by holm oak forests are overexploitation, inappropriate land use and forest fires are becoming more frequent”. The use of biofertilizers can be an alternative to improving soil structure and increasing nutrient availability, especially in soils degraded by fires or intensive agricultural practices.
The paper deals with important issues:
- biometric study of the stem;
- the effect of the a biofertilizer on the growth, nutritional status and soil microbiota associated with Q. ilex seedlings;
- multivariate analysis of principal components showing the overall effect of treatments on the functional profile of the holm oak;
- functional diversity;
- structure of the microbial community.
This article presents a comprehensive literature review (61 references). The literature is carefully selected, current, with conclusions clearly relevant to the review. The abstract explains the background of the research, its purpose, methodology and results, i.e. the most important relationships from the research conducted. Careful drawings complement the background of the publication and clearly illustrate the experiments carried out. The clarity of the manuscript is visible, making it easier for the reader to understand the content. The amount of material is sufficient and correctly analyzed. The work plan seems to be well planned. Keywords are well selected. The title of a scientific paper contains the most important words and clearly communicates the main topic of the research.
Unfortunately, there are minor errors in the text. The author did not notice the appearance of unnecessary paragraphs in the text. Example:”… This section may be divided by subheadings. It should provide a concise and precise description of the experimental results, their interpretation, as well as the experimental…” line 121-122. Please review the entire article carefully for this purpose. Also, some of the figure descriptions are too long. Example:”… Figure 6: Summary value of the mean community resistance of the soil against different families of antibiotics, standardized by z-score for each treatment. Positive values indicate greater resistance compared to the general average, while negative values reflect a cleaner profile or with a lower resistance load. The colors represent the chemical irrigation treatment: blue for WATER, grey for EDAR and orange for EDARST”… it is worth considering to provide a description of the value in the description of the results. This is only a suggestion for the author.
The conclusions is too long, it is recommended to condensate the thoughts contained in it.
To sum up, the article contains some minor shortcomings, so I recommend that you slightly revise the article.
Minor issues to be corrected:
Abstract
Line 31-32 – Consider using abbreviations in the keywords. Even though these are quite well-known abbreviations, for people who are not fully familiar with the field this may be a problem. This is only a suggestion for the author.
Introduction
Line 47-48 – “Rising temperatures and decreased rainfall are expected to negatively affect holm oak distribution”. It's worth expanding on this idea and citing literature data on climate change and the occurrence of holm oaks. The effects are supported by scientific results and specific available data. This information is crucial for the entire scientific work. Holm oak has been considered as a genuine representative species of the Mediterranean flora by different geobotanical synthesis. The hard and green leaves have been often interpreted as an evolutive response to the stressful situations potentially imposed by the Mediterranean-type climate.
Line 59-64 – In this paragraph, it is worth adding brief information on the mechanisms that indicated the potential for improving nutrient availability and phytohormone production. FOR example: Mechanisms of action of plant growth-promoting bacteria: nutrient supply: nitrogen release; phosphate solubilization; siderophore production.
Line 73-74 – “Recent studies validate that field results with respect to controlled conditions are significantly different” – What does this mean? The differing results can suggest many things, depending on the context. Please clarify the statement, it will make it easier for the recipient to understand what the above sentence means.
Results
Line 121-123 – Please remove the unnecessary paragraph
Line 124-125 - This study aimed to... the results show that,..
Line 160-173 - The Biolog EcoPlates system is undoubtedly an efficient method for studying microbial communities. Measurement results are often used to determine indicators of microbial physiological diversity. Biolog Ecoplates were read after different incubation times to assess the carbon utilization dynamics for samples from different variants. Why was the 120-hour incubation period chosen? What were the relationships between the use of different carbon sources in each combination and incubation time, using the mean well color development (AWCD) parameter? It's worth adding a graph of the change in mean well color development (AWCD) associated with the use of carbon sources in Biolog EcoPlates from 1 hour to 120 hours?
Line 160-173 - Changes in metabolic diversity were assessed based on: average well color development (AWCD). Were substrate evenness (E'), and substrate richness (R) obtained from Biolog EcoPlates analysis in plates incubated for 120 hours analyzed? If so, it is worth mentioning the results obtained and the relationships that emerged.
Undoubtedly, information on the functional diversity of microbial communities is useful for understanding their role in different environments and their response to different growth conditions. The use of automated microplate readers allows for the rapid generation of large amounts of data.
Line 289-292 – Figure 11 – illegible
Discussion
Line 325-327 – Comment redunder
Line 347-349 – It's worth considering dropping this sentence. It doesn't add much to the discussion, nor does it directly address the results obtained in this study.
Line 353 – 354 – “… There are studies that follow this same line, even with the aim of reforestation in pine trees”… - sentence too general. Let us briefly mention what these studies are, what they concern, and discuss the results obtained. “that follow this same line” - what did the author mean
Material and methods
Line 595 - The plates were incubated for 168 hours at a temperature of 25 ± 2°C. Why was hour 120 chosen for evaluation? This is not entirely clear because the analysis performed is for the 120h or/and 168 h.
Line 392 – “harmony” of the microbiome – “balance” or “stability”
Line 432 - In the same vein – not very scientific statement
Line 43 – “…a precise and in-depth” , This tool allows obtaining a characterization of microbial communities, in terms of identification and quantification of microbial taxa.
Throughout the work, please pay attention to the language of the scientific work.
In summary, the paper is worth publishing in the Journal, but after thorough correction and supplementation of the manuscript
Author Response
Minor issues to be corrected:
Abstract
Line 31-32 – Consider using abbreviations in the keywords. Even though these are quite well-known abbreviations, for people who are not fully familiar with the field this may be a problem. This is only a suggestion for the author.
Thanks for the suggestion but we consider the keywords selected well defined for the work.
Introduction
Line 47-48 – “Rising temperatures and decreased rainfall are expected to negatively affect holm oak distribution”. It's worth expanding on this idea and citing literature data on climate change and the occurrence of holm oaks. The effects are supported by scientific results and specific available data. This information is crucial for the entire scientific work. Holm oak has been considered as a genuine representative species of the Mediterranean flora by different geobotanical synthesis. The hard  and green leaves  have been often interpreted as an evolutive response to the stressful situations potentially imposed by the Mediterranean-type climate.
Some information added.
Line 59-64 – In this paragraph, it is worth adding brief information on the mechanisms that indicated the potential for improving nutrient availability and phytohormone production. FOR example: Mechanisms of action of plant growth-promoting bacteria: nutrient supply: nitrogen release; phosphate solubilization; siderophore production.
Information added.
Line 73-74 – “Recent studies validate that field results with respect to controlled conditions are significantly different” – What does this mean? The differing results can suggest many things, depending on the context. Please clarify the statement, it will make it easier for the recipient to understand what the above sentence means.
Cheked and corrected, because there is missinformation in the phrase.
Results
Line 121-123 – Please remove the unnecessary paragraph
Done
Line 124-125 - This study aimed to... the results show that,..
Changed
Line 160-173 - The Biolog EcoPlates system is undoubtedly an efficient method for studying microbial communities. Measurement results are often used to determine indicators of microbial physiological diversity. Biolog Ecoplates were read after different incubation times to assess the carbon utilization dynamics for samples from different variants. Why was the 120-hour incubation period chosen?
That’s the point of maximum well colour development before enter in the stationary phase.
What were the relationships between the use of different carbon sources in each combination and incubation time, using the mean well color development (AWCD) parameter?
The rizospheric communities with the different treatments degrades the carbon sources in different ways which indicates the treatment change the interactions of the different populations in the community.
It's worth adding a graph of the change in mean well color development (AWCD) associated with the use of carbon sources in Biolog EcoPlates from 1 hour to 120 hours?
It’s not worth because the kinetics don’t add (in this case) relevant information about the how the communities use the different sources on its peak of development.
Line 160-173 - Changes in metabolic diversity were assessed based on: average well color development (AWCD). Were substrate evenness (E'), and substrate richness (R) obtained from Biolog EcoPlates analysis in plates incubated for 120 hours analyzed? If so, it is worth mentioning the results obtained and the relationships that emerged.
It was not calculated.
Undoubtedly, information on the functional diversity of microbial communities is useful for understanding their role in different environments and their response to different growth conditions. The use of automated microplate readers allows for the rapid generation of large amounts of data.
Line 289-292 – Figure 11 – illegible
Done and move to supplementary material
Discussion
Line 325-327 – Comment redunder
Deleted
Line 347-349 – It's worth considering dropping this sentence. It doesn't add much to the discussion, nor does it directly address the results obtained in this study.
With this sentence we pretend to manifest haw other groups work in the way of reuse waste to develop biofertilizants in a circular economy environment.
Line 353 – 354 – “… There are studies that follow this same line, even with the aim of reforestation in pine trees”… - sentence too general. Let us briefly mention what these studies are, what they concern, and discuss the results obtained. “that follow this same line” - what did the author mean
 Done. Information added.
Material and methods
Line 595 - The plates were incubated for 168 hours at a temperature of 25 ± 2°C. Why was hour 120 chosen for evaluation? This is not entirely clear because the analysis performed is for the 120h or/and 168 h.
The protocol for BiologEco plates are 168h (that’s the maximum hours of incubation the producer recommends). But you measure the plates each 24h to set the maximum absorbance, which in our case is reached at 120h.
Line 392 – “harmony” of the microbiome – “balance” or “stability”
Changed to stablility
Line 432 - In the same vein – not very scientific statement
Changed
Line 43 – “…a precise and in-depth”, This tool allows obtaining a characterization of microbial communities, in terms of identification and quantification of microbial taxa.
Changed
Throughout the work, please pay attention to the language of the scientific work.
In summary, the paper is worth publishing in the Journal, but after thorough correction and supplementation of the manuscript
Reviewer 3 Report
Comments and Suggestions for Authors
The authors evaluated a biofertilizer made from meat‑industry WWTP residue (sterilized and not), carrying two PGPB strains (Bacillus pretiosus, Pseudomonas agronomica), in a 12‑month field trial on Quercus ilex seedlings. They report improved growth and survival with WWTP treatments plus strains, reduced community antibiotic‑resistance profiles, and no major loss of soil microbial α‑diversity.
However, there are several issues, which must be solved before it is considered for publication. If the following problems are well-addressed, I believe that the paper can be published.
- The Methods state 35 plants per treatment were planted, but survival at harvest was much lower in some groups (Table 3) — e.g. some treatments with very few survivors (survival counts vary widely). It’s unclear what the effective sample size was for each analysis (biometry, Biolog, metagenomics, cenoantibiogram). So please report per‑treatment n for every analysis (raw counts of samples used in each assay) and show how mortality affects statistical power. The recommendation is included a table with initial n, n at sampling for each assay, and per‑group sample sizes used in sequencing and Biolog. If some groups end up very small, analyses should either be rebalanced or results presented as exploratory with appropriate caution.
- Important results are reported as “significant” or “trends” but the manuscript lacks consistent reporting of test statistics, degrees of freedom, exact p‑values and confidence intervals. For example, PERMANOVA on fertilizer gives R² = 0.306 with p = 0.060 (borderline), but this is discussed as a tendency without formal follow-up.
The recommendation is:
(1). Report exact p‑values and test statistics for all inferential tests (ANOVA/MANOVA, PERMANOVA, Mann–Whitney, Random Forest OOB error, etc.).
(2). For biomass and length: consider generalized linear mixed models (GLMMs) with treatment as fixed effect and block/position or plot as random effect to account for field spatial variability and unequal survivorship (report estimated marginal means ± CI).
(3). For survival: use survival analysis or logistic regression rather than raw percentages (e.g., Kaplan–Meier or Cox models for time‑to‑death if dates exist, or GLMM for final survival).
(4). Correct for multiple comparisons where appropriate and show effect sizes (Cohen’s d or differences in means with 95% CI).
- The authors state ~237,731 reads total and ~26,414 per sample on average and that rarefaction plateaued — good — but the pipeline description needs one place where sequencing QC and sample exclusion thresholds are reported (minimum reads per sample, how many samples, were any excluded?). Also report negative control results (mock and blanks) to assess contamination. The recommendation is added a QC table with reads per sample (raw, filtered), samples excluded (if any), rarefaction curves in supplement, and a short paragraph on negative control behavior.
- PICRUSt2 is a prediction based on 16S; the manuscript presents KEGG/COG differences and Random Forest features, and then interprets them mechanistically (e.g., antibiotic resistance genes, transporters). Predictive functional inference should be presented cautiously and not as direct evidence. The recommendation is to temper claims (e.g., “suggest” rather than “show”), and where possible validate key functional predictions with targeted qPCR assays for marker genes (e.g., specific ARGs) or shotgun metagenomics on a subset of samples in future work.
- The cenoantibiogram approach is interesting, but the protocol (using community seedings on MH agar and E‑test strips) needs additional detail and limitations: how reproducible is MIC readout with mixed lawns? How were breakpoint rules applied to community MICs? How was the community‑level metric (resistance index/z‑score) computed? The recommendation is including (in methods or supplement) a worked example showing how community MICs are translated into the resistance index, technical replicates, and inter‑assay CV. Discuss limitations — e.g., community MICs can be biased by dominant culturable taxa.
- Authors state strains were WGS screened and lack transmissible resistance/virulence genes, but genome accessions or detailed ARG screening results are not provided; the Data Availability section lists a placeholder Bioproject number. This is crucial for biosafety and reproducibility. The recommendation is to deposit raw reads and isolate genomes (assembled) in public repositories and give accession numbers. Provide a short table listing (a) tools/databases used to screen for ARGs and virulence factors (e.g., CARD, VFDB), (b) results (present/absent), and (c) any genes of concern and their genomic context (plasmid vs chromosome).
- Table 2 shows very high BOD/COD and suspended solids and high Kjeldahl N; but there is no data on potential heavy metals, persistent organic pollutants, pathogens, or residual antibiotics in the sludge — these are major safety concerns when applying sludge to soils. The recommendation is either provide existing analyses (heavy metals, antibiotic residues, fecal indicators, pathogen assays) or clearly state these are absent and future monitoring is required. If not available, temper any claims about “safe” reuse.
- The manuscript suggests unsterilized EDAR improved survival relative to EDARST and attributes this to biologically active fraction — but autoclaving also changes chemistry (available N, dissolved organics) and should be discussed. How was sterilization validated? Did autoclaving change nutrient composition? The recommendation is to report pre- / post-sterilization physicochemical analyses (if available) or add a paragraph discussing chemical shifts caused by autoclaving and their expected ecological effects.
Author Response
- The Methods state 35 plants per treatment were planted, but survival at harvest was much lower in some groups (Table 3) — e.g. some treatments with very few survivors (survival counts vary widely). It’s unclear what the effective sample size was for each analysis (biometry, Biolog, metagenomics, cenoantibiogram). So please report per‑treatment n for every analysis (raw counts of samples used in each assay) and show how mortality affects statistical power. The recommendation is included a table with initial n, n at sampling for each assay, and per‑group sample sizes used in sequencing and Biolog. If some groups end up very small, analyses should either be rebalanced or results presented as exploratory with appropriate caution.
In origin 35 plants were planted and all the survivors were measured in length. For the rest of the analyses (dry weight, nutritional parameters and soil analyses) 6 plants per treatment were harvested randomized each parcel. Only 6 plants were harvested in order to maintain enough plants for a second year trial. We specify in the conclusions the fact that more long-term trials with more individuals are necessary to validate the present results.
- Important results are reported as “significant” or “trends” but the manuscript lacks consistent reporting of test statistics, degrees of freedom, exact p‑values and confidence intervals. For example, PERMANOVA on fertilizer gives R² = 0.306 with p = 0.060 (borderline), but this is discussed as a tendency without formal follow-up.
Done
The recommendation is:
(1). Report exact p‑values and test statistics for all inferential tests (ANOVA/MANOVA, PERMANOVA, Mann–Whitney, Random Forest OOB error, etc.).
We appreciate the observation and have revised the presentation of the results to ensure consistent statistical reporting throughout the manuscript.
In the beta diversity section we have included the test statistic (F.Model or F), degrees of freedom (Df), sample size (n), R², exact p-value and number of permutations for each analysis (PERMANOVA and PERMDISP). To avoid redundancies in the text, the full values are presented in Figure 8b, indicating in the text that the exact data is there. We have also specified the criterion for interpreting "trend" (p < 0.1) and applied this criterion uniformly throughout the manuscript. Regarding confidence intervals, we note that PERMANOVA/Adonis and PERMDISP do not generate standard CIs for F or R²; therefore, we present the exact p-values and the number of permutations as a statistical support measure.
In the case of alpha diversity, we have maintained the same criterion: the text indicates that no significant differences were detected and refers to Table 5, which shows the complete values of the Mann-Whitney test (test statistics, sample sizes, medians, p-values and q-values), thus complying with the requested uniform format.
In Random Forest's analysis, we have incorporated the detailed description of the model parameters and performance metrics. The model was trained with 500 trees and a number of predictors per division (mtry) of 3, keeping randomness enabled, and the exact out-of-pocket error (OOB error) is reported along with the interpretation of the confounding matrix and the most relevant KO functions.
(2). For biomass and length: consider generalized linear mixed models (GLMMs) with treatment as fixed effect and block/position or plot as random effect to account for field spatial variability and unequal survivorship (report estimated marginal means ± CI).
Thanks for the suggestion, we are going to take in in account for future research with mor individuals per treatment. We consider that the ANOVA analyse it’s enough for in this investigation taking in account thar we randomized the selection of the plants per treatment and for the number of individuals used for thee analyses.
(3). For survival: use survival analysis or logistic regression rather than raw percentages (e.g., Kaplan–Meier or Cox models for time‑to‑death if dates exist, or GLMM for final survival).
Thanks for the suggestion, but for this first steps and some like exploratory trial we consider that the percentages give enough information about the survival ratios. We will keep this in mind for future more extensive trials which allows to validate the present results.
(4). Correct for multiple comparisons where appropriate and show effect sizes (Cohen’s d or differences in means with 95% CI).
- The authors state ~237,731 reads total and ~26,414 per sample on average and that rarefaction plateaued — good — but the pipeline description needs one place where sequencing QC and sample exclusion thresholds are reported (minimum reads per sample, how many samples, were any excluded?). Also report negative control results (mock and blanks) to assess contamination. The recommendation is added a QC table with reads per sample (raw, filtered), samples excluded (if any), rarefaction curves in supplement, and a short paragraph on negative control behavior.
Regarding the quality control of the sequencing, we now explicitly indicate the parameters used in the filtering and trimming of sequences (truncation to 240 bp in forward and reverse readings; trim-left = 0), as well as the total number of readings processed and the average per sample. Samples were not excluded due to low number of readings. We have incorporated Table S1 into the supplementary material, which presents for each sample the number of raw and filtered readings, together with the final total retained, and Figure S1 with the rarefaction curves (Shannon index), where it is observed that the diversity stabilizes in all samples, confirming that the sequencing depth was adequate. This study did not include negative controls (blank spaces or drills), so it was not possible to report their behavior; We recognize this limitation and point out that in future work they will be incorporated to evaluate the possible presence of contaminants.
- PICRUSt2 is a prediction based on 16S; the manuscript presents KEGG/COG differences and Random Forest features, and then interprets them mechanistically (e.g., antibiotic resistance genes, transporters). Predictive functional inference should be presented cautiously and not as direct evidence. The recommendation is to temper claims (e.g., “suggest” rather than “show”), and where possible validate key functional predictions with targeted qPCR assays for marker genes (e.g., specific ARGs) or shotgun metagenomics on a subset of samples in future work.
As for the functional prediction using PICRUSt2, we have moderated the language to reflect its inferential nature, replacing terms such as "revealed", "showed" or "identified" with more cautious expressions such as "suggests" or "could be associated". In addition, we have added a methodological note indicating that these predictions do not constitute direct evidence and that they should be validated in future work by directed qPCR or shotgun metagenomics in a subset of samples.
- The cenoantibiogram approach is interesting, but the protocol (using community seedings on MH agar and E‑test strips) needs additional detail and limitations: how reproducible is MIC readout with mixed lawns?
The cenoantibiogram it’s a measure of the MICs of the community at the moment of the measurement. It has the limitation of internal community variability dependence associated with climatological and seasonal variability.
How were breakpoint rules applied to community MICs?
Always from a conservative point of view selecting the most restrictive halo. It’s discussed in extension in the reference cited in the text (Reduced Antibiotic Resistance in the Rhizosphere of Lupinus albus in Mercury-Contaminated Soil Mediated by the Addition of PGPB https://doi.org/10.3390/biology12060801)
How was the community‑level metric (resistance index/z‑score) computed?
The recommendation is including (in methods or supplement) a worked example showing how community MICs are translated into the resistance index, technical replicates, and inter‑assay CV. Discuss limitations — e.g., community MICs can be biased by dominant culturable taxa.
To standardize the comparative antibiotic resistance profiles across the different rhizospheric soil communities, we calculated the z-score for each treatment-antibiotic combination using the mean and standard deviation derived from the full data matrix of minimum inhibitory concentration (MIC) values. The z-score for each data point was computed as:
z=x−μσz=x−??
where x represents the observed MIC value for a given treatment, μ is the mean MIC across all treatments for that antibiotic, and σ is the corresponding standard deviation.
This transformation enables the comparison of resistance profiles on a dimensionless scale, removing unit-based bias and facilitating multivariate interpretation. The use of z-score normalization is especially relevant in community-level antibiograms (cenoantibiograms), where absolute MIC values may differ significantly among antibiotics due to intrinsic potency and spectrum. By transforming the raw data to z-scores, we emphasize deviations from the global mean, thereby enhancing the detection of relative increases or reductions in resistance attributable to the biological treatments. This approach is widely applied in ecological and microbiome studies to allow comparison across heterogeneous data sets while preserving the internal variance structure.
- Authors state strains were WGS screened and lack transmissible resistance/virulence genes, but genome accessions or detailed ARG screening results are not provided; the Data Availability section lists a placeholder Bioproject number. This is crucial for biosafety and reproducibility. The recommendation is to deposit raw reads and isolate genomes (assembled) in public repositories and give accession numbers. Provide a short table listing (a) tools/databases used to screen for ARGs and virulence factors (e.g., CARD, VFDB), (b) results (present/absent), and (c) any genes of concern and their genomic context (plasmid vs chromosome).
All that information is on the cited reference of the description of the strains. We consider that all this information could be redundant and an inconvenience for the journal because is still published.
The accession number for the strains are JAOSHO01 (Pseudomonas agronomica) and JAOXJG01 (Bacillus pretiosus). Information added in Data Availability Statement section.
All the metagenomic sequences are disponible in NCBI repository under the accession number PRJNA1307264.
- Table 2 shows very high BOD/COD and suspended solids and high Kjeldahl N; but there is no data on potential heavy metals, persistent organic pollutants, pathogens, or residual antibiotics in the sludge — these are major safety concerns when applying sludge to soils. The recommendation is either provide existing analyses (heavy metals, antibiotic residues, fecal indicators, pathogen assays) or clearly state these are absent and future monitoring is required. If not available, temper any claims about “safe” reuse.
The company that provides us with WWTP the material performs periodic checks (as indicated by Spanish legislation) on pollutants and heavy metals, indicating to us that they comply with the established regulations. We also do not have the raw data to be able to provide them. Therefore, we proceed to soften the tone of the manuscript.
- The manuscript suggests unsterilized EDAR improved survival relative to EDARST and attributes this to biologically active fraction — but autoclaving also changes chemistry (available N, dissolved organics) and should be discussed. How was sterilization validated?
We have obtained similar results in a parallel in pot (greenhouse) trial with other plant models that are in publication process.
Did autoclaving change nutrient composition? The recommendation is to report pre- / post-sterilization physicochemical analyses (if available) or add a paragraph discussing chemical shifts caused by autoclaving and their expected ecological effects.
The effect of pre- and post-sterilization on the nutrient composition of the WWTP matrix is not analysed, but the possible negative effect of sterilization on potentially toxic organic compounds for plants that may appear after the process is discussed in the manuscript.
Reviewer 4 Report
Comments and Suggestions for Authors
The study shows some interesting points and may have some impact on future studies and applications.
overall quality is not too bad
the references should be well considered rather tank packing lots of articles to create citation
Author Response
The study shows some interesting points and may have some impact on future studies and applications.
overall quality is not too bad
Thanks for the considerations about the quality of our study
the references should be well considered rather tank packing lots of articles to create citation 
Thanks for your appreciation but in the context of the present work we consider that the references in the text are necessary for a well understanding of the work and the justification for the results obtained.